# Semiquantitative Real-Time PCR to Distinguish *Pneumocystis* Pneumonia from Colonization in a Heterogeneous Population of HIV-Negative Immunocompromised Patients

Stine Grønseth,[a] Tormod Rogne,[b,i] Raisa Hannula,[c] Bjørn Olav Åsvold,[d,e,f] Jan Egil Afset,[a,g] Jan Kristian Damås[a,c,h]

[a]Department of Clinical and Molecular Medicine, NTNU, Trondheim, Norway
[b]Department of Circulation and Medical Imaging, NTNU, Trondheim, Norway
[c]Department of Infectious Diseases, St. Olavs hospital, Trondheim University Hospital, Trondheim, Norway
[d]K.G. Jebsen Center for Genetic Epidemiology, Department of Public Health and Nursing, NTNU, Trondheim, Norway
[e]HUNT Research Center, Department of Public Health and Nursing, NTNU, Levanger, Norway
[f]Department of Endocrinology, St. Olavs hospital, Trondheim University Hospital, Trondheim, Norway
[g]Department of Medical Microbiology, St. Olavs hospital, Trondheim University Hospital, Trondheim, Norway
[h]Centre of Molecular Inflammation Research, NTNU, Trondheim, Norway
[i]Department of Chronic Disease Epidemiology, Center for Perinatal, Pediatric and Environmental Epidemiology, Yale School of Public Health, New Haven, Connecticut, USA

**ABSTRACT** *Pneumocystis jirovecii* is a threat to iatrogenically immunosuppressed individuals, a heterogeneous population at rapid growth. We assessed the ability of an in-house semiquantitative real-time PCR assay to discriminate *Pneumocystis* pneumonia (PCP) from colonization and identified risk factors for infection in these patients. Retrospectively, 242 PCR-positive patients were compared according to PCP status, including strata by immunosuppressive conditions, human immunodeficiency virus (HIV) infection excluded. Associations between host characteristics and cycle threshold ($C_T$) values, semiquantitative real-time PCR correlates of fungal loads in lower respiratory tract specimens, were investigated. $C_T$ values differed significantly according to PCP status. Overall, a $C_T$ value of 36 allowed differentiation between PCP and colonization with sensitivity and specificity of 71.3% and 77.1%, respectively. A $C_T$ value of less than 31 confirmed PCP, whereas no $C_T$ value permitted exclusion. A considerable diversity was uncovered; solid organ transplant (SOT) recipients had significantly higher fungal loads than patients with hematological malignancies. In SOT recipients, a $C_T$ cutoff value of 36 resulted in sensitivity and specificity of 95.0% and 83.3%, respectively. In patients with hematological malignancies, a higher $C_T$ cutoff value of 37 improved sensitivity to 88.5% but reduced specificity to 66.7%. For other conditions, assay validity appeared inferior. Corticosteroid usage was an independent predictor of PCP in a multivariable analysis and was associated with higher fungal loads at PCP expression. Semiquantitative real-time PCR improves differentiation between PCP and colonization in immunocompromised HIV-negative individuals with acute respiratory syndromes. However, heterogeneity in disease evolution requires separate cutoff values across intrinsic and iatrogenic predisposition for predicting non-HIV PCP.

**IMPORTANCE** *Pneumocystis jirovecii* is potentially life threatening to an increasing number of individuals with compromised immune systems. This microorganism can cause severe pneumonia in susceptible hosts, including patients with cancer and autoimmune diseases and people undergoing solid organ transplantation. Together, these patients constitute an ever-diverse population. In this paper, we demonstrate that the heterogeneity herein has important implications for how we diagnose and assess the risk of *Pneumocystis* pneumonia (PCP). Specifically, low loads of microorganisms are sufficient

Address correspondence to Stine Grønseth, stine.gronseth@ntnu.no.

Semiquantitative real-time PCR can improve differentiation between non-HIV PCP and colonization, but a significant heterogeneity in fungal loads at disease evolution requires separate cut-off values across non-HIV immunosuppressive predispositions.

to cause infection in patients with blood cancer compared to those in solid organ recipients. With this new insight into host versus *P. jirovecii* biology, clinicians can manage patients at risk of PCP more accurately. As a result, we take a significant step toward offering precision medicine to a vulnerable patient population. One the one hand, these patients have propensity for adverse effects from antimicrobial treatment. On the other hand, this population is susceptible to life-threatening infections, including PCP.

**KEYWORDS** *Pneumocystis jirovecii*, PCP, colonization, immunosuppression, real-time PCR

*P*neumocystis jirovecii is an atypical fungus and causative agent of *Pneumocystis* pneumonia (PCP) (1). Historically, PCP reemerged with the onset of the human immunodeficiency virus (HIV) epidemic as an opportunistic infection and hallmark of AIDS in the 1980s (2). Since the introduction of antiretroviral therapy and prompt administration of PCP prophylaxis, this disease burden is declining (3). Rather, it is becoming overshadowed by PCP in non-HIV immunocompromised populations, especially in resource-rich countries with universal health care (3). Nowadays, *P. jirovecii* represents a life threat to patients with malignancies, immunological disorders, chronic lung diseases, and those undergoing solid organ transplantation (SOT) (4). Their susceptibility to PCP is largely attributed to iatrogenic immunosuppression besides intrinsic host factors (5).

The clinical characteristics of PCP vary according to the degree of immunosuppression and, more markedly, with respect to the host's HIV status (3). First, non-HIV patients typically have a more fulminant onset, rapid progression of severe pneumonitis with respiratory failure, and higher mortality (4). Second, their respiratory samples contain fewer *P. jirovecii* organisms and more neutrophils, features of both diagnostic and prognostic importance (1). Although HIV status is the principal host distinction, HIV-negative patients represent a heterogeneous population with diverse risk profiles (3). Moreover, diagnosing non-HIV PCP is notoriously difficult due to absence of pathognomonic features and a broad differential (6).

Diagnostic guidelines for PCP recommend a multimodal algorithm including detection of *P. jirovecii* (7). Microscopic visualization has been the gold standard, since culturing of *P. jirovecii* is extremely difficult, but the sensitivity of microscopy is especially poor when applied to respiratory samples from non-HIV patients (1). Since the 1990s, highly sensitive PCR-based assays have become widely utilized (8). However, difficulties with differentiating between PCP and colonization, that is, presence of *P. jirovecii* in the absence of clinical pneumonia, has proven a drawback of this technology (4). In fact, this has repercussions for antimicrobial treatment guidance. Prompt initiation is vital for the prognosis of PCP, whereas management of colonization remains debated (1). Our objective was to assess the utility of an in-house semiquantitative real-time PCR-assay for diagnosing PCP in HIV-negative immunocompromised patients and identify predictors for infection.

## RESULTS

**Description of study population and comparisons according to PCP status.** A total of 242 HIV-negative patients (100 female, 142 male) with positive *P. jirovecii* PCR were included, representing 84.0% of 288 presumed eligible patients (Fig. 1). Patient characteristics and univariate comparison according to PCP status are presented in Table 1.

With the present case definition, the condition was classified as PCP (PCP+) in 196 patients and as colonization (PCP−) in 46 patients. Demographics were comparable apart from cardiovascular comorbidity being more common among PCP− patients. Chronic lung diseases were associated with colonization. Otherwise, PCP status seemed independent of immunosuppressive condition and regimen. However, the median corticosteroid dose (first quartile [$q_1$] to third quartile [$q_3$]) at presentation was higher among PCP+ patients (10 [5 to 24] versus 4 [4 to 8] mg methylprednisolone/

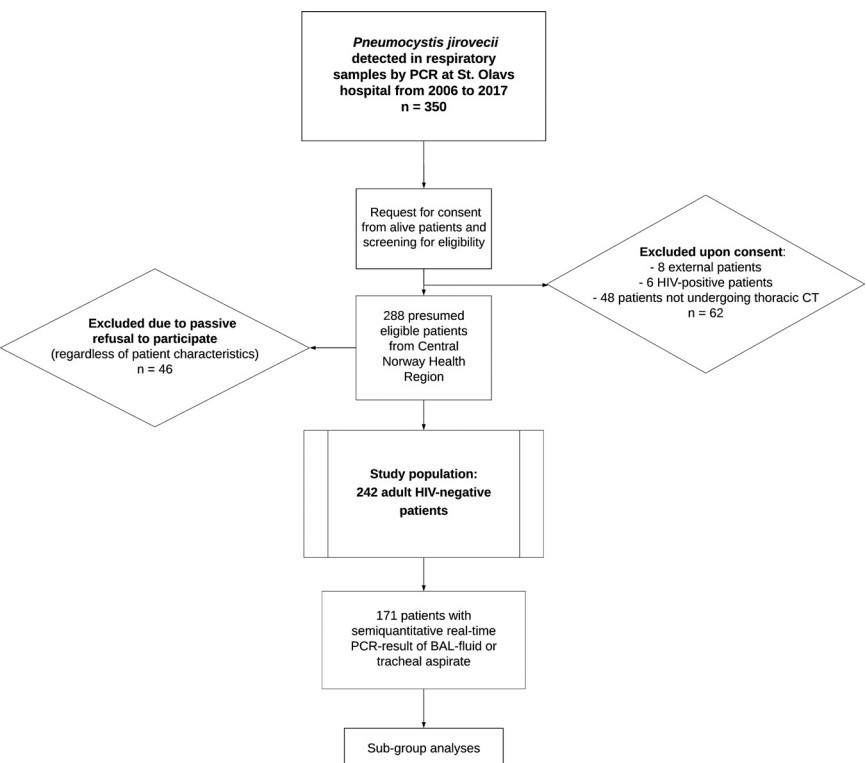

**FIG 1** Flowchart of the study population. Adult patients tested in the regional referral laboratory and undergoing thoracic CT during diagnostic workup were eligible for inclusion. External referral and HIV seropositivity were exclusion criteria. All deceased patients were included, whereas recruitment of alive patients required active consent. BAL, bronchoalveolar lavage; CT, computed tomography; HIV, human immunodeficiency virus; PCP, *Pneumocystis* pneumonia.

day, $P < 0.001$). Besides, PCP$^+$ patients manifested more signs and symptoms of respiratory impairment and specific laboratory and radiological abnormalities (e.g., lymphopenia and crazy paving, respectively).

**Sensitivity of microscopy and diagnostic discrimination by semiquantitative real-time PCR.** Respiratory samples were mainly collected as bronchoalveolar lavage (BAL) fluid ($n = 203$, 83.9%), followed by sputum ($n = 25$, 10.3%), induced sputum ($n = 8$, 3.3%), tracheal aspirate ($n = 4$, 1.7%), respiratory biopsy sample ($n = 1$, 0.4%), and nasopharyngeal swab sample ($n = 1$, 0.4%) (see Fig. S1 in the supplemental material). Direct immunofluorescence (DIF) microscopy was performed on 99 samples, with 44 (44.4%) examinations resulting in positives. The sensitivity of DIF microscopy for *P. jirovecii* detection was positively associated with low cycle threshold ($C_T$) values, regardless of respiratory sample (adjusted odds ratio [OR], 0.77; 95% confidence interval [CI], 0.66 to 0.89) (Fig. S2).

$C_T$ values from semiquantitative real-time PCR analysis of BAL fluid or tracheal aspirate were retrievable for 171 patients (Table S5). The median ($q_1$ to $q_3$) $C_T$ value was lower among PCP$^+$ patients than among PCP$^-$ patients (35 [32 to 37] versus 38 [37 to 41], $P < 0.001$) (Fig. S3), confirming higher fungal loads in individuals with clinical infection. However, it was impossible to find an optimal $C_T$ cutoff value for discrimination between PCP and colonization due to overlaps (Fig. S4). The receiver operating characteristic (ROC) curve analysis gave an area under the curve (AUC) of 0.80 (95% CI, 0.73 to 0.88) (Fig. 2A). A $C_T$ value of 36 came closest to maximizing sensitivity and specificity simultaneously, being 71.3% (95% CI, 63.7% to 78.9%) and 77.1% (95% CI, 63.2% to 91.1%), respectively. This corresponded to a positive predictive value of 92.4% (95% CI, 87.3% to 97.5%) and a negative predictive value of 40.9% (95% CI, 29.0 to 52.8%). The validity and percentage of correctly classified patients varied according to $C_T$ cutoff

**TABLE 1** Characteristics of study population and comparison of patients with *Pneumocystis* pneumonia and colonization[a]

| Characteristic | No. (%) in case of missing | Value Study population (n = 242; 100%) | PCP+ (n = 196; 81.0%) | PCP− (n = 46; 19.0%) | P value difference |
|---|---|---|---|---|---|
| **Demographics and comorbidity** | | | | | |
| Median age (yrs [$q_1$–$q_3$]) | NA | 66 (59–73) | 65.5 (59–73) | 68 (60–74) | 0.39 |
| Male sex (no. [%]) | NA | 142 (58.7) | 119 (60.7) | 23 (50.0) | 0.18 |
| History of smoking (no. [%]) | 235 (97.1) | 131 (55.7) | 106 (55.8) | 25 (55.6) | 0.98 |
| Median Charlson comorbidity index ($q_1$–$q_3$) | NA | 6 (4–8) | 6 (4–8) | 6 (4–8) | 0.97 |
| Comorbidities (no. [%]) | NA | | | | |
| Cardiovascular disease | | 66 (27.3) | 45 (23.0) | 21 (45.7) | 0.002 |
| Chronic kidney disease | | 32 (13.2) | 26 (13.3) | 6 (13.0) | 0.97 |
| Chronic liver disease | | 2 (0.83) | 2 (1.0) | 0 (0.0) | 1.00 |
| Chronic pulmonary disease | | 43 (17.8) | 32 (16.3) | 11 (23.9) | 0.23 |
| Congestive heart failure | | 13 (5.4) | 10 (5.1) | 3 (6.5) | 0.72 |
| Diabetes mellitus type 1 or 2 | | 33 (13.6) | 26 (13.3) | 7 (15.2) | 0.73 |
| Hematological malignancy[b] | | 12 (5.0) | 10 (5.1) | 2 (4.3) | 1.00 |
| Hypertension | | 75 (31.0) | 60 (32.1) | 15 (27.3) | 0.79 |
| Rheumatic disease | | 7 (2.9) | 6 (3.1) | 1 (2.2) | 1.00 |
| Solid tumor | | 28 (11.6) | 24 (12.2) | 4 (8.7) | 0.62 |
| Any of the above | | 157 (64.9) | 124 (63.3) | 33 (71.7) | 0.28 |
| Primary PCP prophylaxis at presentation | NA | 2 (0.8) | 2 (1.0) | 0 (0) | 1.00 |
| **Microbiology** | | | | | |
| $C_T$ value of semiquantitative real-time PCR-analysis (median [$q_1$–$q_3$]) | | | | | |
| Any respiratory sample[c] | 202 (83.5) | 36 (33 to 37) | 35 (32–37) | 38 (37–41) | <0.001 |
| BAL fluid or tracheal aspirate[c] | 171 (70.7) | 36 (33–37) | 35 (32–37) | 38 (37–41) | <0.001 |
| **Immunosuppressive conditions** | | | | | |
| Distribution across PCP groups | NA | | | | 0.19 |
| Hematological malignancies | | 89 (37.6) | 75 (38.3) | 14 (30.4) | Ref. |
| Solid tumors | | 68 (28.7) | 59 (30.1) | 9 (19.6) | 0.66 |
| Immunological disorders | | 38 (16.0) | 28 (14.3) | 10 (21.7) | 0.17 |
| Solid organ transplantation | | 29 (12.2) | 23 (11.7) | 6 (13.0) | 0.54 |
| Chronic lung diseases | | 13 (5.5) | 8 (4.1) | 5 (10.9) | 0.059 |
| Other/miscellaneous[d] | | 5 (2.1) | 3 (1.5) | 2 (4.3) | Excluded |
| Pulmonary metastasis from solid tumor | | 12 (5.0) | 9 (4.6) | 3 (6.5) | 0.70 |
| **Premorbid iatrogenic immunosuppression, chemotherapy and corticosteroid exposure** | | | | | |
| Any immunosuppressive regimen (no. [%]) | NA | | | | |
| Last 5 yrs | | 230 (95.0) | 187 (95.4) | 43 (93.5) | 0.70 |
| At presentation | | 205 (84.7) | 168 (85.7) | 37 (80.4) | 0.37 |
| Regimen at presentation (no. [%]) | NA | | | | 0.33 |
| Chemotherapy for hematological malignancy and adjuvant steroids | | 54 (22.3) | 47 (24.0) | 7 (15.2) | |
| Chemotherapy for solid tumor and adjuvant steroids | | 31 (12.8) | 26 (13.3) | 5 (10.9) | |
| Chemotherapy for hematological malignancy | | 10 (4.1) | 10 (5.1) | 0 (0) | |
| Chemotherapy for solid tumor | | 14 (5.8) | 11 (5.6) | 3 (6.5) | |
| Corticosteroids in monotherapy | | 35 (14.5) | 29 (14.8) | 6 (13.0) | |

**TABLE 1** (Continued)

| Characteristic | No. (%) in case of missing | Value Study population (n = 242; 100%) | PCP+ (n = 196; 81.0%) | PCP− (n = 46; 19.0%) | P value difference |
|---|---|---|---|---|---|
| Graft rejection prophylaxis after SOT | | | | | |
| DMARDs with or without adjunctive steroids | | 28 (11.6) | 23 (11.7) | 5 (10.9) | 0.31 |
| Other combinations[e] | | 22 (9.1) | 15 (7.7) | 7 (15.2) | 0.96 |
| None | | 11 (4.6) | 7 (3.6) | 4 (8.7) | 0.20 |
| | | 37 (15.3) | 28 (14.3) | 9 (19.6) | Ref. |
| Systemic corticosteroid exposure pattern 60 days preceding presentation (no. [%]) | 240 (99.2) | | | | <0.001 |
| Daily | | 102 (42.5) | 80 (41.2) | 22 (47.8) | |
| Intermittent | | 74 (30.8) | 64 (33.0) | 10 (21.7) | |
| None | | 64 (26.7) | 50 (25.8) | 14 (30.4) | |
| Methylprednisolone equivalent dose (mg/day at presentation) (median [$q_1$–$q_3$])[f] | 237 (97.9) | 8 (4–20) | 10 (5–24) | 4 (4–8) | |
| Symptomatology (no. [%]) | | | | | |
| Cough | NA | 140 (57.9) | 117 (59.7) | 23 (50.0)) | 0.23 |
| Dyspnea | NA | 184 (76.0) | 156 (79.6) | 37 (60.9) | 0.007 |
| Fever | NA | 180 (74.4) | 151 (77.0) | 29 (63.0) | 0.05 |
| Minimum two cardinal symptoms | NA | 184 (76.0) | 154 (78.6) | 30 (65.2) | 0.056 |
| All three cardinal symptoms | NA | 81 (33.5) | 74 (37.8) | 7 (15.2) | 0.004 |
| No cardinal symptoms | NA | 3 (1.2) | 0 (0) | 3 (6.5) | 0.007 |
| Objective findings and biochemistry | | | | | |
| Abnormal lung auscultation (no. [%]) | NA | 144 (59.5) | 123 (62.8) | 21 (45.7.) | 0.033 |
| Oxygen saturation (%) (median [$q_1$–$q_3$])[g] | 207 (85.5) | 89 (84–93) | 88 (84–93) | 91.5 (88–95) | 0.014 |
| Leukocyte count × 10⁹/liter (median [$q_1$–$q_3$]) | 235 (97.1) | 7.0 (4.3–10) | 6.9 (4.2–10.0) | 7.7 (5.2–9.9) | 0.36 |
| Neutrophil count × 10⁹/liter (median [$q_1$–$q_3$]) | 186 (76.9) | 4.8 (2.8–7.3) | 4.8 (2.8–7.3) | 4.8 (3.1–7.0) | 0.99 |
| Neutropenia (<0.5 neutrophils 10⁹/liter) | 186 (76.9) | 3 (1.6) | 2 (1.3) | 1 (3.6) | 0.37 |
| Lymphocyte count × 10⁹/liter (median [$q_1$–$q_3$])[h] | 122 (50.4) | 0.63 (0.41–1.1) | 0.6 (0.4–1.1) | 1.0 (0.5–1.5) | 0.047 |
| Lymphopenia (<1.0 lymphocyte × 10⁹/liter) | 123 (50.8) | 82 (66.7) | 73 (70.2) | 9 (47.4) | 0.052 |
| CD4+ T cell count × 10⁹/liter (median [$q_1$–$q_3$]) | 13 (5.4) | 0.13 (0.07–0.25) | 0.1 (0.05–0.25) | 0.32 (0.22–0.41) | 0.24 |
| Lactate dehydrogenase (U/liter) (median [$q_1$–$q_3$]) | 142 (58.7) | 293.5 (221–390) | 308 (225–390) | 224 (200–441) | 0.082 |
| Albumin (g/liter) (median [$q_1$–$q_3$]) | 174 (71.9) | 33 (27–36) | 32.5 (27–35.5) | 33.5 (27–37.5) | 0.95 |
| C-reactive protein (mg/liter) (median [$q_1$–$q_3$]) | 235 (97.1) | 76 (38–146) | 81 (42–156) | 53 (24.5–116.5) | 0.019 |
| Radiological features (no. [%]) | | | | | |
| Any remarks on chest X-ray | 204 (84.3) | 160 (78.4) | 133 (80.1) | 27 (71.1) | 0.22 |
| Any remarks on thoracic CT | NA | 237 (97.9) | 196 (100) | 41 (89.1) | <0.001 |
| Findings on thoracic CT | NA | | | | |
| Atelectasis | | 41 (16.9) | 29 (14.8) | 12 (26.1)) | 0.066 |
| Bronchiectasis | | 18 (7.4) | 11 (5.6) | 7 (15.2) | 0.025 |
| Crazy paving pattern | | 55 (22.3) | 53 (27.0) | 4 8.7) | 0.007 |
| Consolidations | | 44 (18.2) | 39 (19.9) | 5 (10.9) | 0.20 |
| Cysts | | 9 (3.7) | 6 (3.1) | 3 (6.5) | 0.38 |
| Emphysema | | 26 (10.7) | 20 (10.2) | 6 (13.0) | 0.58 |
| Ground glass opacities[i] | | 180 (74.4) | 171 (87.2) | 12 (26.1) | <0.001 |
| Infiltrates[j] | | 52 (21.5) | 42 (21.4) | 10 (21.7) | 0.96 |
| Lymphadenopathy | | 40 (16.5) | 32 (16.3) | 8 (17.4) | 0.86 |

**TABLE 1** (Continued)

| Characteristic | No. (%) in case of missing | Value | | | P value difference |
| --- | --- | --- | --- | --- | --- |
| | | Study population (n = 242; 100%) | PCP⁺ (n = 196; 81.0%) | PCP⁻ (n = 46; 19.0%) | |
| Noduli | | 21 (8.7) | 15 (7.7) | 6 (13.0) | 0.24 |
| Pleural effusion | | 67 (27.7) | 52 (26.5) | 15 (32.6) | 0.41 |
| Pneumothorax | | 1 (0.41) | 1 (0.5) | 0 (0.0) | 1.00 |
| Reticular or septal thickening | | 63 (26.0) | 55 (28.1) | 8 (17.4) | 0.14 |
| "Tree-in-bud sign" | | 16 (6.6) | 11 (5.3) | 5 (10.9) | 0.20 |

[a]Criteria for PCP were multimodal and based on available patient data (see Materials and Methods and Fig. S1 in the supplemental material). Patients not fulfilling the criteria for their respective groups were considered colonized with *P. jirovecii* (i.e., PCP⁻). BAL, bronchoalveolar lavage; CT, computed tomography; $C_T$, cycle threshold; DMARDs, disease-modifying antirheumatic drugs; NA, not applicable; Ref, reference group in logistic regression analysis; SOT, solid organ transplantation.

[b]In 12 patients, hematological malignancy was not considered the primary immunosuppressive condition or an indication for immunosuppression but rather a comorbidity.

[c]Respiratory samples included bronchoalveolar lavage fluid (n = 203, 83.9%), induced sputum (n = 25, 10.3%), sputum (n = 8, 3.3%), tracheal aspirate (n = 4, 1.7%), respiratory biopsy specimen (n = 1, 0.4%) and nasopharyngeal swab sample (n = 1, 0.4%), in a total of 242 samples. $C_T$ values were retrievable from analysis of 202 samples, including 171 BAL fluid samples and tracheal aspirates.

[d]Other/miscellaneous immunosuppressive conditions included two patients with no diagnosed condition, whereas two had received steroids for suspected autoimmune disorder and one patient with statin-induced myositis was treated with corticosteroids.

[e]Other combinations include exposure to other immunosuppressants (mycophenolate, azathioprine, cyclophosphamide, calcineurin and mTOR inhibitors, and cyclosporine and hydroxychloroquine with or without adjuvant steroids) and one patient receiving both graft rejection prophylaxis for solid organ transplantation and chemotherapy for hematological malignancy with adjuvant corticosteroids.

[f]Median methylprednisolone equivalent dose per day was calculated among 117 patients having an intake the day of *P. jirovecii* detection: 95 PCP⁺ and 22 PCP⁻ patients.

[g]Fifty-three patients were receiving supplemental oxygen when saturation was measured; 45 (23.0%) in the PCP⁺ group and 8 (17.4%) in the PCP⁻ group (P = 0.41 for difference).

[h]One patient with chronic lymphatic leukemia was excluded from the analysis due to an abnormally high lymphocyte count (i.e., 37.9 × 10⁹/liter).

[i]Note: Ground glass opacities and infiltrates were among the criteria for PCP⁺.

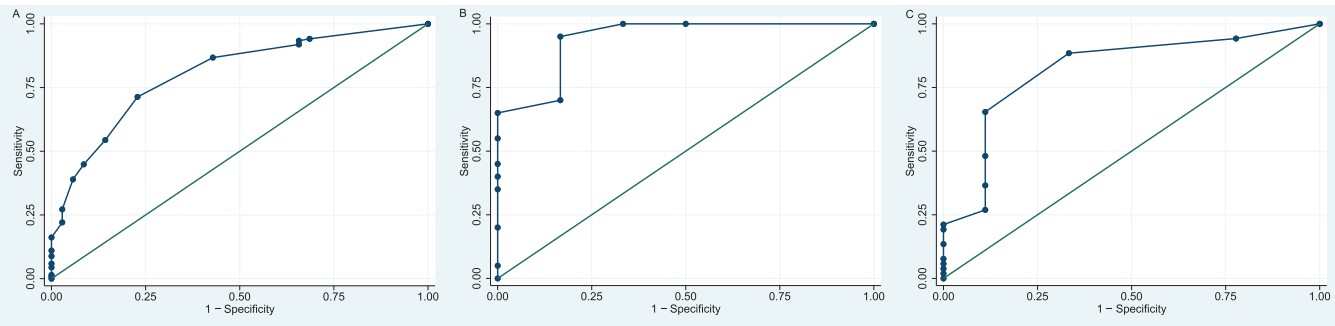

**FIG 2** ROC curves of semiquantitative real-time PCR of BAL fluid or tracheal aspirate for discrimination between *Pneumocystis* pneumonia and colonization. (A) ROC curve for population overall, based on 171 samples. (B) ROC curve for SOT recipients, based on 26 samples. (C) ROC curve for patients with hematological malignancies, based on 61 samples.

value (Fig. S5). $C_T$ values greater than 36 defined a gray zone without definitive discrimination, comprising 39 PCP$^+$ patients. Their characteristics are summarized in Table S2.

**Subgroup analyses of PCP$^+$ patients.** $C_T$ values of <31 corresponded to 100% specificity. To identify characteristics of this subpopulation with higher fungal loads ($n = 22$), we compared it to PCP$^+$ patients with $C_T$ values of 31 and higher ($n = 114$) (Table S3). Notably, fungal load appeared associated with immunosuppressive condition ($P = 0.05$). SOT recipients accounted for 36.4% of the high-fungal-load population, whereas patients with hematological malignancies dominated the low-fungal-load population, constituting 40.5%. Moreover, we noted an association between corticosteroid exposure and fungal burden, with more daily users and fewer unexposed subjects in the high-fungal-load population. Median doses were comparable.

**Heterogeneity in fungal loads.** Successively, we further analyzed the relationships to immunosuppressive predisposition, including corticosteroid exposure and fungal burden (Fig. 3; see also Fig. S6 and S7). A linear regression model was fitted comparing $C_T$ values in BAL fluid or tracheal aspirate across immunosuppressive conditions, with patients with hematological malignancies as a reference group, (F[4,162] = 3.03, $P = 0.019$, $R^2 = 0.070$). Only SOT recipients had significantly lower $C_T$ values (Table S4). Univariate analyses confirmed this difference in medians ($q_1$ to $q_3$) compared to patients with hematological malignancies overall (34.5 [28 to 36] versus 36 [34 to 37], $P = 0.072$), among PCP$^+$ patients (33 [28 to 36] versus 36 [33 to 37], $P < 0.01$), and to a lesser degree among PCP$^-$ patients (38 [37 to 38] versus 39.5 [37 to 41], $P = 0.54$).

**Discrimination across immunosuppressive conditions.** With caution regarding the number of patients and observations, we investigated the validity of semiquantitative real-time PCR across immunosuppressive conditions. Based on 26 samples from SOT recipients, the discrimination between PCP and colonization appeared outstanding and superior to the population overall (AUC, 0.94; 95% CI, 0.82 to 1.00) (Fig. 2B). A $C_T$ value of 36 corresponded to a sensitivity of 95.0% (95% CI, 85.4% to 100.0%) and a specificity of 83.3% (95% CI, 53.5% to 100.0%). In spite of lower fungal loads, the validity was excellent for patients with hematological malignancies (AUC, 0.82; 95% CI, 0.66 to 0.98) based on 61 observations (Fig. 2C). Yet, a higher $C_T$ cutoff value was needed to achieve a sensitivity of >75%. Here, a $C_T$ value of 37 corresponded to a sensitivity of 88.5% (95% CI, 79.8% to 97.1%) and a specificity of 66.7% (95% CI, 35.9% to 97.5%). The validity of the PCR assay appeared inferior for the remaining conditions (Fig. S8A to C; Table S5).

**Independent risk factors for PCP.** Based on univariate comparisons, we performed multivariable analyses to identify independent risk factors for PCP (Table 2). Only chronic lung diseases were associated with markedly lower odds of PCP (OR, 0.30; 95% CI, 0.09 to 1.05). Presence of all three cardinal symptoms and abnormal lung auscultation were independent predictors for PCP. Moreover, corticosteroid dose at presentation was positively associated with PCP, while $C_T$ value and oxygen saturation were negative predictors. The presence of crazy paving on computed tomography (CT) imaging was strongly associated with PCP.

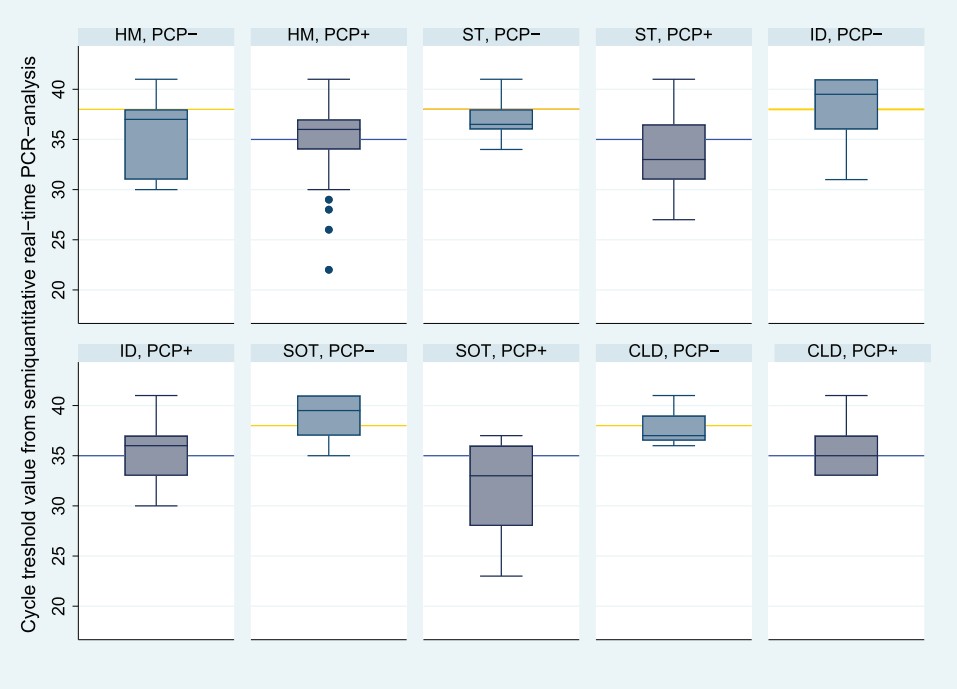

**FIG 3** Relationship between semiquantitative real-time PCR-result, immunosuppressive conditions, and PCP status. $C_T$ values from of BAL fluid or tracheal aspirate differed significantly according to PCP status ($P < 0.01$) with medians being 35 (blue line) and 38 (yellow line), respectively. Retrospectively, 196 patients were diagnosed with PCP (i.e., PCP$^+$) while 46 were presumed colonized (i.e., PCP$^-$). $C_T$, cycle threshold; CLD, chronic lung disease; HM, hematological malignancy; ID, immunological disorder; PCP, *Pneumocystis* pneumonia; PCR, polymerase chain reaction, SOT, solid organ transplant; ST, solid tumor.

## DISCUSSION

This study demonstrates that semiquantitative real-time PCR can improve differentiation between PCP and colonization in immunocompromised HIV-negative patients. However, a significant heterogeneity in fungal loads across immunosuppressive predispositions implicates that universal cutoff values for predicting non-HIV PCP are inadequate.

Non-HIV populations at risk of opportunistic infections, including PCP, are growing rapidly because of prolonged survival and escalating use of immunosuppressants (3, 5). Diagnostic algorithms with high specificity are needed to avoid unnecessary treatment, especially among multimorbid patients with propensity for adverse effects and drug interactions (9). On the other hand, delayed diagnosis is associated with increased mortality risk, potentially exceeding 50% (1).

Semiquantitative real-time PCR gradually substituted microscopy for *P. jirovecii* detection in our regional referral laboratory during the last decades, but whether $C_T$ values should be emphasized for treatment guidance remained unestablished. Here, the study subjects represented a selected population, and they had high pretest probability of PCP. Accordingly, the majority were classified as PCP$^+$ in retrospect. Although $C_T$ values were significantly lower among PCP$^+$ patients, it was impossible to determine a cutoff with a 100% negative predictive value.

Several studies have assessed real-time PCR strategies to distinguish PCP from colonization. Extrapolation is limited by heterogeneity in PCR targets, PCP definitions, host characteristics, types of respiratory samples, sample volumes, DNA extraction, and quantification methods ($C_T$ values or copies per milliliter) (10). Anyhow, the majority have found real-time PCR assays potentially useful (11–25), though gray zones are common and stratification by HIV status is of utmost importance. Inability to

**TABLE 2** Uni- and multivariable analyses of risk factors for *Pneumocystis* pneumonia versus colonization[a]

| Risk factor and covariate(s)[b] | No. of observations | OR[d] | 95% CI | P value |
|---|---|---|---|---|
| **Cardiovascular comorbidity** | **NA** | **0.35** | **0.18–0.69** | **0.002** |
| Age and sex | NA | 0.27 | 0.13–0.57 | 0.002 |
| Any other comorbidity and sex | NA | 0.29 | 0.14–0.60 | 0.001 |
| Daily methylprednisolone equivalent dose at presentation among exposed/mg increase | 117 | 0.47 | 0.17–1.26 | 0.13 |
| **C$_T$ value of semiquantitative real-time PCR-analysis of BAL fluid or tracheal aspirate/unit increase** | **171** | **0.68** | **0.58–0.80** | **<0.001** |
| Daily methylprednisolone equivalent dose at presentation among exposed/mg increase | 82 | 0.54 | 0.38–0.80 | < 0.001 |
| **Immunosuppressive condition**[c] | **237** | | | |
| **Hepatological malignancy** | **89** | **1** | **Ref.** | **Ref.** |
| **Solid tumor** | **68** | **1.22** | **0.50–3.02** | **0.66** |
| **Immunological disorder** | **38** | **0.52** | **0.21–1.31** | **0.17** |
| **Solid organ transplantation** | **29** | **0.72** | **0.25–2.07** | **0.54** |
| **Chronic lung disease** | **13** | **0.30** | **0.09–1.05** | **0.059** |
| **Daily methylprednisolone equivalent dose at presentation/mg increase** | **237** | **1.05** | **1.00–1.10** | **0.035** |
| **Daily methylprednisolone equivalent dose at presentation among exposed/mg increase** | **117** | **1.11** | **1.02–1.20** | **0.011** |
| **Dyspnea** | **242** | **2.51** | **1.26–4.98** | **0.009** |
| Cardiovascular comorbidity | 242 | 2.87 | 1.30–5.88 | 0.004 |
| Immunosuppressive condition | 237 | 2.83 | 1.36–5.89 | 0.005 |
| Systemic corticosteroid exposure pattern 60 days preceding presentation | 240 | 2.84 | 1.40–5.46 | 0.004 |
| **Fever** | **242** | **1.97** | **0.99–3.90** | **0.053** |
| Daily methylprednisolone equivalent dose at presentation/mg increase | 237 | 2.33 | 1.14–4.75 | 0.020 |
| Daily methylprednisolone equivalent dose at presentation among exposed/mg increase | 117 | 2.68 | 0.96–7.45 | 0.059 |
| **At least two cardinal symptoms (cough, dyspnea, fever)** | **242** | **1.96** | **0.97–3.92** | **0.059** |
| Immunosuppressive condition | 237 | 1.70 | 0.81–3.55 | 0.159 |
| Daily methylprednisolone equivalent dose at presentation among exposed/mg increase | 117 | 2.52 | 0.92–6.93 | 0.073 |
| **All three cardinal symptoms (cough, dyspnea, and fever)** | **242** | **3.38** | **1.44–7.94** | **0.005** |
| Daily methylprednisolone equivalent dose at presentation/mg increase | 237 | 4.28 | 1.71–10.7 | 0.002 |
| Daily methylprednisolone equivalent dose at presentation among exposed/mg increase | 117 | 6.23 | 1.30–29.7 | 0.022 |
| **Abnormal lung auscultation** | **242** | **2.01** | **1.05–3.84** | **0.035** |
| Daily methylprednisolone equivalent dose at presentation/mg increase | 237 | 1.81 | 0.93–3.51 | 0.080 |
| Daily methylprednisolone equivalent dose at presentation among exposed/mg increase | 117 | 3.35 | 1.22–9.21 | 0.019 |
| Immunosuppressive regimen at presentation | 242 | 2.17 | 1.10–4.28 | 0.026 |
| **Oxygen saturation in %/unit increase** | **207** | **0.93** | **0.87–0.99** | **0.016** |
| **Lymphocyte count × 10⁹/liter/unit increase** | **122** | **0.71** | **0.50–1.00** | **0.050** |
| Daily methylprednisolone equivalent dose at presentation among exposed/mg increase | 59 | 1.13 | 0.55–2.32 | 0.745 |
| Immunosuppressive condition[c] | 119 | 0.64 | 0.43–0.94 | 0.024 |
| **Lymphopenia (<1.0 × 10⁹/liter)** | **123** | **2.62** | **0.97–7.07** | **0.058** |
| Charlson comorbidity index/unit increase | 123 | 2.97 | 1.06–8.32 | 0.039 |
| Daily methylprednisolone equivalent dose at presentation/mg increase | 120 | 2.87 | 1.04–7.92 | 0.042 |

**TABLE 2** (Continued)

| Risk factor and covariate(s)[b] | No. of observations | OR[d] | 95% CI | P value |
|---|---|---|---|---|
| Daily methylprednisolone equivalent dose at presentation among exposed/mg increase | 60 | 2.35 | 0.56–9.94 | 0.244 |
| **C-reactive protein in mg/liter/unit increase** | **235** | **1.00** | **1.00–1.01** | **0.057** |
| **Lactate dehydrogenase in U/liter/unit increase** | **142** | **1.00** | **1.00–1.00** | **0.89** |
| **Atelectasis** | **242** | **0.49** | **0.23–1.06** | **0.070** |
| Daily methylprednisolone equivalent dose at presentation among exposed/mg increase | 117 | 0.70 | 0.22–2.21 | 0.54 |
| Lymphocyte count × 10⁹/liter/unit increase | 122 | 2.86 | 0.35–23.2 | 0.33 |
| Immunosuppressive regimen at presentation | 242 | 0.57 | 0.26–1.25 | 0.16 |
| **Bronchiectasis** | **242** | **0.33** | **0.12–0.91** | **0.032** |
| Age, sex | 242 | 0.37 | 0.13–1.05 | 0.063 |
| Immunosuppressive condition[c] | 237 | 0.43 | 0.15–1.27 | 0.13 |
| Systemic corticosteroid exposure pattern 60 days preceding presentation | 240 | 0.37 | 0.13–1.02 | 0.054 |
| Daily methylprednisolone equivalent dose at presentation among exposed/mg increase | 237 | 0.52 | 0.11–2.41 | 0.40 |
| Immunosuppressive regimen at presentation | 242 | 0.37 | 0.13–1.1 | 0.073 |
| **Crazy paving pattern on thoracic CT** | **242** | **3.89** | **1.33–11.4** | **0.013** |
| Age and sex | 242 | 4.28 | 1.45–12.7 | 0.009 |
| $C_T$ value of semiquantitative real-time PCR analysis of BAL fluid or tracheal aspirate | 171 | 6.09 | 1.58–23.4 | 0.009 |
| Immunosuppressive condition[c] | 237 | 4.38 | 1.45–13.3 | 0.009 |
| Immunosuppressive regimen at presentation | 242 | 4.29 | 1.44–12.8 | 0.009 |
| Daily methylprednisolone equivalent dose at presentation among exposed/mg increase | 117 | 5.26 | 1.12–24.8 | 0.036 |
| Lymphocyte count × 10⁹/liter/unit increase | 122 | 3.07 | 0.65–14.4 | 0.16 |

[a]Criteria for PCP were multimodal and based on available patient data (see Materials and Methods and Fig. S1). Patients not fulfilling the criteria for their respective groups were considered colonized with *P. jirovecii* (i.e., PCP⁻). BAL, bronchoalveolar lavage; CT, computed tomography; $C_T$, cycle threshold; NA, not applicable; OR, odds ratio.

[b]Risk factors are in boldface. Plausible confounders were identified *a priori* and included in multivariable analyses. Covariates with ≥10% effect on OR are included in the table. For complete list of covariates, refer to Table S1.

[c]Five patients had immunosuppressive conditions classified as miscellaneous and were excluded from the comparative analysis. Adjustment for age and sex did not cause significant changes to odds ratios overall or *P* values and are not reported.

[d]Univariate analysis results are in boldface; adjusted ORs are in lightface.

discriminate the two entities has also been described (26, 27), perhaps due to a continuous progression from carriage to active infection (7).

Upon exposure, *P. jirovecii* adheres to type 1 pneumocytes, which in turn induces organism activation and multiplication (1). The passage from colonization to PCP and complications is ill defined in non-HIV patients (5), and CD4 counts fail in predicting disease (6). Paradoxically, the associated lung injury is proposed to result from an inappropriate inflammatory host response (5). Marked bronchoalveolar neutrophilia observed in HIV-negative patients likely reflects this reaction and aggravates the prognosis (1).

Since the fungus lives and thrives in the alveoli, an increasing density gradient from the upper to the lower respiratory tract is expected (7). In the attempt to avoid invasive sampling, researchers have assessed the validity of upper respiratory tract specimens compared to the gold standard of BAL fluid, with various results (10). Overall, the sensitivity appears too low to exclude PCP, while positive results support the diagnosis (7). Asymptomatic carriage in the upper respiratory tract due to recent exposure is a differential diagnosis (4), and a theoretical source of contamination unless protective invasive sampling is applied (28).

In light of the current knowledge gaps and diagnostic challenges, a major strength of this study is the large number of high-risk cases and high-yield respiratory specimens permitting subgroup analyses. Interestingly, SOT recipients and patients with hematological malignancies distinguished themselves at different ends of a spectrum, harboring high and low fungal loads, respectively. However, an $R^2$ of 7.0% suggests that endogenous host predisposition explains little of the diversity. Indeed, our results indicate that immunosuppression, including corticosteroid exposure, also influences the precise intersection of host response and *P. jirovecii* concentration that results in clinical infection.

Cancer patients are primarily subject to cycles of chemotherapy regimens, for instance, rituximab, cyclophosphamide, vincristine, and prednisolone (R-CHOP) and fludarabine, cyclosporine, and rituximab (FCR), both involving significant risk of PCP (5). Moreover, corticosteroids have vast supportive care indications in oncology, increasing exposure (2). In comparison, SOT recipients are prescribed daily multidrug regimens with explicit lymphocytotoxic effects to prevent allograft rejection (29). Although SOT regimens are pleiotropic and not CD4 specific, perhaps they come closest to mimicking the lymphocyte depletion occurring during the natural course of HIV infection considering their continuity and intensity (29).

Notably, Montesinos et al. found that *P. jirovecii* concentrations were markedly heterogeneous in samples from HIV-negative PCP patients (23). Relatedly, Robert-Gangneux et al. highlighted hematological malignancies particularly for the tendency of negative microscopy examinations, *per se*, to be associated with low fungal loads (26). Altogether, we hypothesize that intrinsic and iatrogenic host factors affect *P. jirovecii* multiplication and non-HIV PCP expression. Regardless of the pathogenesis, our findings have important implications. Foremost, the validity of real-time PCR strategies may vary across immunosuppressive predispositions, and optimal cutoff values for discrimination should be validated according to these strata.

Acknowledging the importance of the recent multicenter study from the Fungal PCR Initiative comparing the performance of several commercial and noncommercial *P. jirovecii* quantitative real-time PCR assays with emphasis on standardization, our in-house assay harbors certain shortcomings (30). Specifically, the protocol only tests the efficacy of the amplification step. Ideally, one should add a negative control prior to extraction to monitor the entire real-time process. Use of an alien negative control is preferable to avoid bias from human factors (e.g., unknown quantity of human DNA in eluate). Moreover, inherent variability of biologic systems is an important bottleneck in real-time PCR studies such as ours. To limit confounding from differences in sample volumes, relative quantification (e.g., the comparative [$\Delta\Delta$] $C_T$ method) involving normalization of *P. jirovecii* to one or more reference genes with near constant expression should prevail over absolute quantification. Importantly, the genes must be amplified

with comparable efficacy for this method to be accurate (31). Owing to higher feasibility, easier clinical interpretation, and determination of cutoff values, diagnostic microbiology departments may still prefer absolute quantification.

The last concern regards the target gene for amplification. Beta-tubulin is a highly conserved single-copy nuclear gene (10). Single-copy genes are favorable to avoid bias in quantification and accurately reflect the quantity of organisms (30). This allows interstrain comparisons and direct determination of cutoff values, since varied copy numbers is a nonissue. However, compared to multicopy gene targets such as the major surface glycoprotein and mitochondrial genes, inferior analytical sensitivity is a drawback (10, 30). Extraction of whole nucleic acids demonstrates an even wider detection range for *P. jirovecii* compared to that with DNA only (30). In fact, to target the mitochondrial small subunit with whole nucleic acid as a starting material appears to yield the best sensitivity (30). The rationale for using assays with the highest sensitivity obtainable is vast. Principally, even low-amount *P. jirovecii* inoculums can be associated with non-HIV PCP. With the distinct exception of SOT patients, our study underscored this characteristic, particularly among patients with hematological malignancies. Hence, the nature of this disease strongly argues for high negative predictive values, including the lower spectrum of *P. jirovecii* inoculums. The growing implications of colonization are equally important. Molecular genotyping reports involving colonized patients in nosocomial transmission networks are worrisome and emphasize the urgency for strategies to reduce circulation of *P. jirovecii* (32). Furthermore, the possible risk of developing full-blown PCP from colonization in case of deteriorated immunity favors preemptive treatment (30).

Despite the above-described issues, we believe that the main findings of our study withstand. Considering the ever-diverse population susceptible to *P. jirovecii*, these indications warrant further investigations with emphasis on appropriate study design and stratified analyses.

Besides real-time PCR, this study underlines readily available clinical characteristics to emphasize for treatment guidance. In line with previous reports (12, 14, 16, 26, 33), the sensitivity of DIF microscopy appeared associated with *P. jirovecii* loads. Concerning noninvasive investigations, history of all three cardinal symptoms and decreased oxygen saturation were independent predictors of PCP in our PCR-positive cohort. Also, lymphopenia, an established risk factor for PCP (5), was associated with PCP, based on 123 observations. In our experience, a common pitfall is declaring patients immunocompetent if their neutrophil count is normal in spite of lymphopenia. In relation to this, cumulative corticosteroid dose is worth stressing due to lymphocytotoxic effects. Although we found a positive association, dose tapering, low doses, or no preceding intake does not exclude PCP (2). Lastly, both corticosteroids and lymphopenia are risk factors for colonization too, complicating clinical discrimination (8).

Cardiovascular comorbidity favored colonization in the univariate analysis. We hypothesize that shared clinical characteristics, particularly in cardiac patients, contributed to this. However, a multivariable analysis confirmed a positive confound by corticosteroids, moderating this relationship. A reluctance toward corticosteroid therapy to these patients because of adverse circulatory and metabolic effects may explain this finding.

This study has several limitations. First, we were unable to include all alive patients. Also, to strive for diagnostic homogeneity, validation of the semiquantitative real-time PCR was primarily performed on lower-respiratory-tract specimens. These limitations represent selection bias. Second, this was a retrospective analysis, challenging data collection and reliability. Third, the lack of a gold standard for diagnosing PCP might have resulted in information bias. Fourth, an increase in familywise error rate across reported statistical analyses was not controlled for. Finally, the comparison of fungal loads is challenged by variability in respiratory specimens, host pathogen biology, and procedural and analytical factors discussed above.

In conclusion, semiquantitative real-time PCR offers high objectivity and sensitivity

for *P. jirovecii* detection in HIV-negative immunocompromised individuals. However, heterogeneity across host predispositions requires multivariable models to optimize discrimination between life-threatening PCP and colonization. Prospective studies are needed to assess the external validity of our results while reducing the risk of bias and confounding.

## MATERIALS AND METHODS

**Setting and inclusion.** St. Olavs hospital, Trondheim University Hospital, is the only tertiary referral hospital in the central Norway health region, covering approximately 700,000 inhabitants. Adult patients with respiratory samples testing positive for *P. jirovecii* by PCR at the Department of Medical Microbiology from 2006 to 2017 were identified. For inclusion, respiratory samples included BAL fluids, induced sputa, sputa, tracheal aspirates, respiratory biopsy specimens, and nasopharyngeal swab samples. Patients who were HIV negative, had been followed up regionally, and had undergone thoracic CT were eligible. Inclusion of alive patients required active consent, while all deceased patients were included.

**Data collection.** Comprehensive biological, clinical, and demographic data were collected retrospectively from patient records. Ongoing corticosteroid intake on the date of *P. jirovecii* detection was registered and converted into the equivalent in methylprednisolone expressed as milligrams per day. Degree of comorbidity was assessed according to the Charlson weighted comorbidity index (34). Cardiovascular comorbidities comprised coronary heart disease, stroke, and peripheral artery disease, whereas congestive heart failure and hypertension were registered separately. Epi Info (version 7.2.2.6; Centers for Disease Control and Prevention, Atlanta, GA, USA) was used for data recording.

**Microbiological detection of *P. jirovecii*.** DIF microscopy was performed with MONOFLUO *Pneumocystis jirovecii* IFA test kit number 32515 (Bio-Rad). Lack of positive controls from "definite" PCP patients was a challenge during the study period. For this reason and concerns regarding sensitivity and specificity, the laboratory used DIF as a complementary method in line with the guidelines (7), mainly on PCR-positive samples. In 2017, semiquantitative real-time PCR replaced DIF definitely. The in-house assay targeting the beta-tubulin gene of *P. jirovecii* was adapted from Brancart et al. (33) with some modifications as described in detail below (11, 33).

**Semiquantitative real-time PCR-protocol.** Respiratory tract samples that were viscous were pretreated with Sputolysin (dithiothreitol, volume 1:2) for 10 min for liquefaction of mucoid fluids before DNA extraction. Next, if the sample volume was >10 ml, 3 to 5 ml was subjected to centrifugation at $3,000 \times g$ for 30 min. Thereafter, 500 $\mu$l of the supernatant was mixed with 50 $\mu$l proteinase K and incubated for 15 min at 65°C. If the sample volume was <10 $\mu$l, the centrifugation step was omitted, and 1 ml of sample was mixed with 100 ml proteinase K and incubated as described above. Then, the mixture was spun down, the supernatant was removed, and 500 $\mu$l of precipitate was used for DNA extraction on a NucliSENS easyMAG instrument (bioMérieux) with an eluate volume of 55 $\mu$l.

Reagents and PCR instruments used varied during the study period, but all changes were validated to ensure equal quality. During the main part of the study period, the following procedure and reagents were used: 5 $\mu$l of eluate was added to 10 $\mu$l of PerfeCTa multiplex qPCR supermix with uracil-*N*-glycosylase, 0.5 $\mu$l of each primer (12 $\mu$M) and probe (8 $\mu$M), and 3.5 $\mu$l molecular-grade water. BAL fluids, considered critical patient samples, were extracted and amplified in duplicates. Amplification reactions were carried out on either a CFX96 real-time system (Bio-Rad), Chromo4 system (Bio-Rad), or LightCycler 2.0 instrument (Roche) with the following cycling conditions: 45°C for 5 min, 95°C for 3 min, and then 40 cycles of 95°C, 60°C, and 72°C for 10 s each. Results were reported to clinicians as negative/positive, with a comment about low concentration of *P. jirovecii* if the cycle threshold ($C_T$) value was high. A cloned PCR product was used as an external positive control, and molecular-grade water was used as a negative control in all PCR runs. To control for inhibition, a separate real-time PCR targeting a human 237-bp intergenic region of chromosome 20 (position 104006 to 104242, sequence AL133466) was run, as previously described (35). All samples were positive, indicating absence of PCR inhibitors, and no samples were excluded due to nonamplification during the study period. The laboratory participated in a *Pneumocystis jirovecii* pneumonia (PCP) DNA EQA Program (QCMD) during the study period.

**Retrieval of $C_T$ values.** $C_T$ values were not reported in the laboratory information system during the study period. Therefore, $C_T$ values were collected from the log of the PCR instruments in retrospect. Since some of the PCR instruments were replaced and discarded during the study period, $C_T$ values for samples run on those instruments were lost. These were registered as "missing" during data collection. The retrievability of $C_T$ values depended on which instrument the analyses were run, and the missing pattern was considered random and unrelated to patient characteristics.

**Case definition.** To separate infection from colonization in PCR-positive patients, multimodal criteria based on current clinical practice, previous reports (36–38), and existing diagnostic guidelines emphasizing biological detection were imposed *a posteriori* (7) (see Fig. S1 in the supplemental material). We identified three patient groups and applied the following criteria for PCP: group 1, (i) immunosuppressive state and (ii) positive DIF; group 2 (characterized by missing or negative DIF microscopy-result), (i) immunosuppressive state, (ii) at least one cardinal symptom of PCP (cough, dyspnea, and fever), (iii) typical findings on thoracic CT (ground glass opacities and/or infiltrates), and (iv) presumptive diagnosis at time of diagnosis, i.e., receiving anti-PCP treatment; group 3, patients who died in hospital within 30 days of detection without receiving anti-PCP treatment. We evaluated these patients individually with respect to cause of death and PCP status to exclude abrupt death from PCP without time to receive

anti-PCP treatment. The alternative diagnosis was colonization and PCP-unrelated death (i.e., terminal patients dying from underlying conditions). Patients not fulfilling the criteria for their respective groups were considered colonized with *P. jirovecii*. $C_T$ values were compared to the retrospective PCP status, infection (PCP$^+$) or colonization (PCP$^-$).

**Statistics.** Continuous and categorical variables are presented as medians with second ($q_1$) and third ($q_3$) quartiles and proportions with percentages, respectively. Simple linear regression was used to compare $C_T$ values across immunosuppressive conditions. Otherwise, univariate analyses were performed with the Wilcoxon rank sum, chi-square, or Fisher's exact test as appropriate, except for polychotomous independent variables, for which logistic regression was applied. Subsequently, multivariable logistic regression analyses were performed for variables having *P* values of <0.10 with covariates identified *a priori* (Table S1), with PCP versus colonization as outcomes. ROC curves were used to assess the validity of semiquantitative real-time PCR and determine sensitivity and specificity according to $C_T$ cutoff values. Results are expressed as proportions, ORs, or AUC with 95% confidence intervals. All *P* values were two sided. Values of <0.05 were considered statistically significant.

Analyses were performed using Microsoft Excel (version 16.4; Microsoft Corporation, Redmond, WA, USA), STATA/MP (version 15.1; StataCorp, College Station, TX, USA), and IBM SPSS statistics for Macintosh (version 27.0; IBM Corp., Armonk NY, USA).

**Ethics.** This study was approved by the Regional Committee for Medical and Health Research Ethics (REC-North, reference number 2017/2419).

## SUPPLEMENTAL MATERIAL

Supplemental material is available online only.

**SUPPLEMENTAL FILE S1**, PDF file, 0.7 MB.

## ACKNOWLEDGMENTS

We thank Andreas Brun and other employees at the Department of Medical Microbiology of St. Olavs hospital for contributing to patient identification and data collection. We also thank Nord-Trøndelag and Møre og Romsdal Hospital Trusts for supporting the project.

This work was supported by the Norwegian Research Council and the Faculty of Medicine and Health Sciences at the Norwegian University of Science and Technology (NTNU) through the participation of S.G. in the Student Research Program at NTNU.

We declare no conflicts of interest.

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
