## [Reviewer comments · Microbiology Spectrum]

**Microbiology
Spectrum**

Semiquantitative real-time PCR to distinguish *Pneumocystis pneumonia* from colonization in a heterogeneous population of HIV-negative immunocompromised patients

Stine Grønseth, Tormod Rogne, Raisa Hannula, Bjørn Olav Åsvold, Jan Egil Afset, and Jan Damås

Corresponding Author(s): Stine Grønseth, Norwegian University of Science and Technology

Review Timeline:

Submission Date:	April 14, 2021
Editorial Decision:	May 18, 2021
Revision Received:	June 10, 2021
Accepted:	June 14, 2021

Editor: Alexandre Alanio

Reviewer(s): The reviewers have opted to remain anonymous.

Transaction Report:

DOI: <https://doi.org/10.1128/Spectrum.00026-21>

May 18, 2021

Ms. Stine Grønseth
Norwegian University of Science and Technology
Department of Clinical and Molecular Medicine
NTNU Department of Clinical and Molecular Medicine
NO-7491
Trondheim
Norway

Re: Spectrum00026-21 (Semiquantitative real-time PCR to distinguish *Pneumocystis* pneumonia from colonization in a heterogenous population of HIV-negative immunocompromised patients)

Dear Ms. Stine Grønseth:

Thank you for submitting your manuscript to Microbiology Spectrum. As you will see the reviewers support publication of a revised paper. Please revise the paper along the lines suggested by the reviewers. When submitting the revised version of your paper, please provide (1) point-by-point responses to the issues raised by the reviewers as file type "Response to Reviewers," not in your cover letter, and (2) a PDF file that indicates the changes from the original submission (by highlighting or underlining the changes) as file type "Marked Up Manuscript - For Review Only". Please use this link to submit your revised manuscript - we strongly recommend that you submit your paper within the next 60 days or reach out to me. Detailed information on submitting your revised paper are below.

Link Not Available

Sincerely,

Alexandre Alanio

Journals Department
Reviewer comments:

Reviewer #2 (Comments for the Author):

The paper by Grønseth is an interesting report that uses semi-quantitative PCR to discriminate between colonization and infection. The paper contains useful data for the field. However, I am not 100% convinced that rat Ct value is the appropriate way to analyze the data. The authors mention using a human target to validate sample integrity (this is what we do for SARS-Cov2 testing as well).

1. What human target was used? Were there any samples that should be thrown out of the analyses due to non-amplification?
2. Would a delta delta Ct, pneumocystis to human target- have more precision?
3. What was the rationale for tubulin as opposed to a mitochondrial DNA or rRNA target that may have greater dynamic range?

Reviewer #3 (Comments for the Author):

The authors describe the assessment of an in-house semiquantitative real-time PCR for the discrimination of *Pneumocystis pneumonia* and colonization. They importantly outline the risk factors of the heterogeneous group of iatrogenically immunocompromised, non-HIV patients. 242 *Pneumocystis*-PCR positive patients were included in this retrospective study which was conducted between 2006 and 2017. The PCP-status stratified by immunosuppressive conditions, associations between host-characteristics and Ct values and fungal loads of lower respiratory tract specimens, were analyzed.

Solid organ transplant patients showed significantly higher fungal loads compared to hematological diseases. Corticosteroid usage was a predictor of PCP and associated with higher fungal loads at PCP-expression.

Compared to several other studies on this topic, this work outlines the facts that the validity of real-time PCR-strategies may vary across immunosuppressive predisposition and stratification may enable to find optimal cut-off values for discrimination. The large number of high-risk cases and respiratory specimen permitted the corresponding sub-group analyses.

The article on this challenging topic is well-written, but needs some clarification before publication.

Major concerns:

Line 171/172 :

Please clarify : « mainly on PCR-positive samples whenever positive controls were available ? »

It seems not logic to perform DIF on PCR-positive samples (Confirmation of PCR with a less sensitive method as microscopy ?)

Line 177 :

Please indicate the active substance of sputolysin and its effect.

Line 191 :

Why only BAL-fluids were extracted and amplified in duplicates? What about the other respiratory samples?

Line 220:

« Patients who died in-hospital within 30 days of detection without receiving anti-PCP treatment». Please clarify, why the patients did not receive a treatment after detection.

Minor concerns:

Line 153 :

Please specify what kind of respiratory samples were tested

Line 183/184/188/192 : Please make sure to use always the correct brand names (NucliSENS easyMAG, PerfeCTa Multiplex qPCR SuperMix, LightCycler 2.0)

Line 197 : « MGW » as negative control. Please explain the abbreviation

Reviewer #4 (Comments for the Author):

The authors used a single nuclear gene as target which is known to be less sensitive. This has already been shown in comparative study to miss low fungal loads of *P. jirovecii*. This should be deeply discussed, as there was an expert consensus to use highly sensitive qPCR methods to detect as much patients as possible with low fungal loads in order to have evidence for treatment or prophylaxis. See Gits Muselli et al. Med Mycol 2019.

MIQE guidelines are recommending Alien internal control and not human for qPCR assays. Human control is a way to evaluate the quality of the sample but the authors cannot control the quantity of human DNA so they cannot know if the whole PCR process (including extraction) went perfectly well. A small decreased performance will not be detected with such procedure (human DNA as a major DNA in eluates will still be amplified). See MIQE guidelines for qPCR assays, please discuss.

Staff Comments:

Preparing Revision Guidelines

- Point-by-point responses to the issues raised by the reviewers in a file named "Response to Reviewers," NOT IN YOUR COVER LETTER.
- Upload a compare copy of the manuscript (without figures) as a "Marked-Up Manuscript" file.
- Each figure must be uploaded as a separate file, and any multipanel figures must be assembled

into one file.

- Manuscript: A .DOC version of the revised manuscript
- Figures: Editable, high-resolution, individual figure files are required at revision, TIFF or EPS files are preferred

For complete guidelines on revision requirements, please see the Instructions to Authors at [link to page]. **Submissions of a paper that does not conform to Microbiology Spectrum guidelines will delay acceptance of your manuscript.**

Please return the manuscript within 60 days; if you cannot complete the modification within this time period, please contact me. If you do not wish to modify the manuscript and prefer to submit it to another journal, please notify me of your decision immediately so that the manuscript may be formally withdrawn from consideration by Microbiology Spectrum.

If you would like to submit an image for consideration as the Featured Image for an issue, please contact Spectrum staff.

Trondheim 10th June 2021

Microbiology Spectrum

Dear Editor,

Thank you for giving us the opportunity to submit a revised draft of our manuscript titled **“Semiquantitative real-time PCR to distinguish *Pneumocystis pneumonia* from colonization in a heterogeneous population of HIV-negative immunocompromised patients”** to include in *Microbiology Spectrum*. We appreciate the time and effort that the reviewers have dedicated to providing valuable feedback. We have incorporated changes to accommodate the raised issues. We have highlighted the changes within the manuscript by using “track changes”.

Please find our point-by-point response to the comments and concerns below. We look forward to hearing from you in due time regarding our submission and to respond to any further questions and comments you may have.

Sincerely,

Stine Grønseth, MD, PhD Candidate

Department of Clinical and Molecular Medicine, Faculty of Medicine and Health Sciences

NTNU - Norwegian University of Science and Technology, Trondheim

Postal address: NTNU Department of Clinical and Molecular Medicine, NO-7491 Trondheim, Norway

Phone: +47-93409532, E-mail: stine.gronseth@ntnu.no

Professor Jan Kristian Damås, MD, PhD

Department of Infectious Diseases, St. Olavs Hospital

Olav Kyrres gt. 17, 7006 Trondheim, Norway

Phone: +47-91112046, E-mail: jan.k.damas@ntnu.no

Point-by-point response to the comments from the reviewers

Reviewer #2:

- *The paper by Grønseth is an interesting report that uses semi-quantitative PCR to discriminate between colonization and infection. The paper contains useful data for the field. However, I am not 100% convinced that rat Ct value is the appropriate way to analyze the data. The authors mention using a human target to validate sample integrity (this is what we do for SARS-Cov2 testing as well).*

Question 1: *What human target was used? Were there any samples that should be thrown out of the analyses due to non-amplification?*

- **Response:** Thank you for raising these questions. The human target used as an internal amplification control was a 237 basepair sequence in an intergenic region in chromosome 20, position 104006-104242 in nucleotide sequence AL133466. In the case of samples with negative results, a comment was added if the human target PCR also were negative, i.e., that the result was inconclusive with a recommendation of repeated testing. During the study period all samples were positive indicating absence of DNA-inhibitors to PCR and no samples were thrown out. We have specified these important issues in the revised manuscript in line with your queries (page 9, line 201-205).
- **Question 2:** *Would a delta delta Ct, pneumocystis to human target- have more precision?*
- **Response:** Thank you for pointing out this highly relevant issue. We fully agree with your concern. Relative quantification is a precise alternative to control for eventual confounding related to inherent biologic variability by normalization to a reference gene. In a research context, this method is preferable. In this study, however, we relied on data from a clinical microbiology department. Here, semi-quantitation of fungal inoculums and C_T values are considered more feasible and less laborious for a diagnostic purpose. In addition, their interpretation in a clinical setting may be more intuitive. Our primary objective was to evaluate whether these C_T values should play a role in treatment guidance. To strive homogeneity in our analyses of C_T values in relation to clinical characteristics, we only included patients with lower respiratory tract samples. Although we can't control for differences in concentrations, we believe that the main findings of our study (e.g., lower C_T values among patients with PCP compared to colonized individuals and heterogeneity in fungal inoculums across non-HIV predispositions) withstand and warrant further investigations. We elaborate on this important topic in the revised discussion (page 19, line 453-459).

- **Question 3:** *What was the rationale for tubulin as opposed to a mitochondrial DNA or rRNA target that may have greater dynamic range?*
- **Response:** Thank you for raising this important question. We fully appreciate your concern. The current assay targeting beta-tubulin was chosen in 2006 for its robustness and clinical utility based on the report from Brancart et al (*Journal of Microbiological Methods*, 2005). The sensitivity and objectivity are superior to those of microscopic examinations, which has been the traditional gold standard for PCP diagnosing. Moreover, the sensitivity limit is low (about 50 copies/mL), the assay is highly reproducible, and the linear range exceeds the minimum standard and covers five orders of magnitude. Lastly, the direct 1:1 organism quantification is considered an important advantage of beta-tubulin and this assay in a clinical context. That said, in terms of dynamic range and sensitivity to detect even very low fungal inoculums, multi-copy genes exceed single-copy genes. Considering the growing implications of colonization (e.g., role in chronic diseases, interhuman transmission networks), and possibility of full-blown PCP developing from colonization or even at very low fungal inoculums, mitigate against beta-tubulin. The last issue was accentuated by patients with hematological malignancies in our study. In parallel with new-gained knowledge, the historical focus on excluding PCP is changing towards increased attention on detecting all individuals infected with *P. jirovecii* with subsequent differentiation between PCP and colonization. This diagnostic shift changes the prerequisites of diagnostic assays like ours. A potential downside of multi-copy genes as targets is variable copy-numbers of the target gene across isolates which may hamper extrapolation, determination of universal cut-off values, and inter-strain comparisons. To target both a single-copy and a multicopy gene could solve this problem. We fully acknowledge all these aspects and have addressed the importance of choosing a suitable target gene in the revised discussion (page 19-20, line 461- 478). At the same time, we believe that the findings from our study are relevant for future optimization of real-time PCR assay with emphasis on the heterogeneity of non-HIV hosts and its implications.

Reviewer #3:

- *The authors describe the assessment of an in-house semiquantitative real-time PCR for the discrimination of *Pneumocystis pneumonia* and colonization. They importantly outline the risk factors of the heterogenous group of iatrogenically immunocompromised non-HIV patients. 242 *Pneumocystis*-PCR positive patients were included in this retrospective study which was conducted between 2006 and 2017. The PCP-status stratified by immunosuppressive conditions, associations between host-characteristics and Ct values and fungal loads of lower respiratory tract specimens, were analyzed. Solid organ transplant patients showed*

significantly higher fungal loads compared to hematological diseases. Corticosteroid usage was a predictor of PCP and associated with higher fungal loads at PCP-expression.

Compared to several other studies on this topic, this work outlines the facts that the validity of real-time PCR-strategies may vary across immunosuppressive predisposition and stratification may enable to find optimal cut-off values for discrimination. The large number of high-risk cases and respiratory specimen permitted the corresponding sub-group analyzes. The article on this challenging topic is well-written, but needs some clarification before publication.

Comment 1: Line 171/172 : *Please clarify: «mainly on PCR-positive samples whenever positive controls were available?» It is seems not logic to perform DIF on PCR-positive samples (Confirmation of PCR with a less sensitive method as microscopy?)*

- **Response:** Thank you for pointing this out. We fully agree with these remarks. DIF was gradually substituted with the PCR-assay in the referral laboratory and 2016 was the last year this assay was performed for PCP diagnosing. During the period DIF was utilized (i.e., 2006-2016), lacking positive controls from “definite” PCP patients was a challenge. For this matter and issues concerning sensitivity and specificity (false negatives and positives, respectively), DIF was used as complementary rather than confirmatory detection method alongside PCR in line the European diagnostic guidelines for non-HIV patients (ECIL 2016; Alanio et al.). We recognize that the original phrasing is susceptible to misinterpretation, and we have revised accordingly.
- **Comment 2:** Line 177: *Please indicate the active substance of sputolysin and its effect.*
- **Response:** Thank for noticing this ambiguity. The active component of Sputolysin is dithiothreitol (DTT) This agent serves for liquefaction of mucoid respiratory specimens before DNA extraction. We have specified this issue, accordingly (page 7-8, line 170-175).
- **Comment 3:** Line 191: *Why only BAL-fluids were extracted and amplified in duplicates? What about the other respiratory samples?*
- **Response:** Thank you for raising this question and underlining this procedure. The rationale is related to the critical nature of the specimen and not the pathogen, *P. jirovecii*. BAL-fluids are extracted and amplified in duplicates as a safety step to avoid invasive re-sampling in case of erroneous handling or testing. Since most of the samples were BAL-fluids, we have little reason to believe that this influenced our results significantly. Moreover, multivariable analyses were restricted to samples from lower respiratory tract to strive homogeneity. To clarify for the readers, we have specified the reason of this procedure in the manuscript (page 8, line 194-195).

- **Comment 4:** Line 220: «Patients who died in-hospital within 30 days of detection without receiving anti-PCP treatment». Please clarify, why the patients did not receive a treatment after detection.
- **Response:** Thank you highlighting this ambiguity. We fully agree that the message is unclear. Regarding the case definition, receipt of PCP-treatment was a PCP case criterion in combination with other clinical criteria (e.g., symptomatology, radiologic findings). To assure that patients who died in proximity to the positive PCR-test without receiving treatment were not missed with this case-definition, we evaluated these patients individually with respect to cause of death and PCP-status. Our main concern was whether their lack of treatment was caused by abrupt death from PCP due to delayed presentation, diagnosis, or treatment. The alternative we considered was PCP-unrelated death, e.g., terminal patients dying from their underlying disease with presumed colonization. This retrospective classification was made based on chart review by an infectious disease specialist. Three patients were evaluated and two were classified as PCP⁺ with this approach (specified in Supplemental Figure 1). We have elaborated and revised the manuscript in line with your comments to clarify this matter for the readers (page 10, line 227-231).
- **Comment 5:** Line 153: Please specify what kind of respiratory samples were tested
- **Response:** Thank you for pointing this out. We have specified the kind of respiratory samples accordingly (page 7, line 153-155).
- **Comment 11:** Line 183/184/188/192: Please make sure to use always the correct brand names (NucliSENS easyMAG, PerfeCTa Multiplex qPCR SuperMix, LightCycler 2.0).
Response: Thank you for this remark. We have corrected the brand names accordingly (page 8, line 189, 192, 193 and 196).
- **Comment 12:** Line 197: «MGW » as negative control. Please explain the abbreviation
- **Response:** Thank you for noticing this imprecision. We have specified accordingly (i.e., Molecular Graded Water (MGW), page 9, line 201).

Reviewer #4:

- **Comment 1:** The authors used a single nuclear gene as target which is known to be less sensitive. This has already been shown in comparative study to miss low fungal loads of *P. jirovecii*. This should be deeply discussed, as there was an expert consensus to use highly sensitive qPCR methods to detect as much patients as possible with low fungal loads in order to have evidence for treatment or prophylaxis. See Gits Muselli et al. Med Mycol 2019.

Response: Thank you for raising this important issue. We fully agree with your remarks. The current assay targeting beta-tubulin was chosen in 2006 for its robustness and clinical utility based on the report from Brancart et al (*Journal of Microbiological Methods*, 2005). The sensitivity and objectivity are superior to those of microscopic examinations, which had been the traditional gold standard for PCP diagnosing. Moreover, the sensitivity limit is low (about 50 copies/mL), the assay is highly reproducible, and the linear range exceeds the minimum standard and covers five orders of magnitude. Lastly, the direct 1:1 organism quantification is considered an important advantage of beta-tubulin and this assay in a clinical context. That said, in terms of dynamic range and sensitivity to detect even very low fungal inoculums, multi-copy genes exceed single-copy genes. Considering the growing implications of colonization (e.g., role in chronic diseases, interhuman transmission networks), and possibility of full-blown PCP developing from colonization or even at very low fungal inoculums, mitigate against beta-tubulin. The last issue was accentuated by patients with hematological malignancies in our study. In parallel with new-gained knowledge, the historical focus on excluding PCP is changing towards increased attention on detecting all individuals infected with *P. jirovecii* with subsequent differentiation between PCP and colonization. This diagnostic shift changes the prerequisites of diagnostic assays like ours. A potential downside of multi-copy genes as targets is variable copy-numbers of the target gene across isolates which may hamper extrapolation, determination of universal cut-off values, and inter-strain comparisons. To target both a single-copy and a multicopy gene could solve this problem. We fully acknowledge all these aspects and have addressed the importance of choosing a suitable target gene in the revised discussion with reference to Gits Muselli et al. *Med Mycol* 2019 (page 19-20, line 461-478). At the same time, we believe that the findings from our study are relevant for future optimization of real-time PCR assay with emphasis on the heterogeneity of non-HIV hosts and its implications.

- **Comment 2:** *MIQE guidelines are recommending Alien internal control and not human for qPCR assays. Human control is a way to evaluate the quality of the sample but the authors cannot control the quantity of human DNA so they cannot know if the whole PCR process (including extraction) went perfectly well. A small decreased performance will not be detected with such procedure (human DNA as a major DNA in eluates will still be amplified). See MIQE guidelines for qPCR assays, please discuss.*
- Thank you for pointing out this highly relevant issue. We completely appreciate your remarks and agree that an alien gene target introduced prior to extraction is the ideal internal control to monitor the entire real-time process without bias from human DNA-concentration in the eluate. With respect to our assay, the internal control is designed as a parallel real-time PCR.

By comparing the C_T values from this analysis to the anticipated results, we get an indication of whether the real-time process has been successful or not, from extraction to amplification. However, as accurately pointed out, the sensitivity of this method to detect extraction errors is not optimal and there is a risk of false negative results related to the extraction process. We have revised the discussion to include these important considerations (page 19, line 449-452).

Additional clarifications and errata

In addition to the above comments, we have corrected spelling and grammatical errors identified during revision. Lastly, we have made the following corrections in wake of review of clinical data revealing that the number of presumed eligible patients were 288 and not 307:

- Figure 1:
 - No. of patients with *P. jirovecii* detected in respiratory sample with PCR at St. Olavs hospital from 2006 to 2017 (from n = 370 to n = 351)
 - No. of presumed eligible patients from Central Norway Health Region (from n = 307 to n = 288)
 - No. of patients excluded patients due to passive refusal to participate (from n = 65 to n = 46)
- Results (page 12, line 276):
 - Percentage of presumed eligible patients included and denominator (from 78.8 % of 307 patients to 84.0 of 288 patients)

June 14, 2021

Ms. Stine Grønseth
Norwegian University of Science and Technology
Department of Clinical and Molecular Medicine
NTNU Department of Clinical and Molecular Medicine
NO-7491
Trondheim
Norway

Re: Spectrum00026-21R1 (Semiquantitative real-time PCR to distinguish *Pneumocystis* pneumonia from colonization in a heterogeneous population of HIV-negative immunocompromised patients)

Dear Ms. Stine Grønseth:

I would like to thank you for your reply to reviewer's comments. The resulting manuscript is better discussing all the technical and clinical issues unresolved or neglected on the subject.

Your manuscript has been accepted, and I am forwarding it to the ASM Journals Department for publication. You will be notified when your proofs are ready to be viewed.

Sincerely,

Alexandre Alanio
Editor, Microbiology Spectrum

Journals Department
Supplemental Dataset: Accept